# Semantic similarity metrics for learned image registration

**Steffen Czolbe**                       PER.SC@DI.KU.DK
**Oswin Krause**                   OSWIN.KRAUSE@DI.KU.DK
*Department of Computer Science, University of Copenhagen, Denmark*

**Aasa Feragen**                       AFHAR@DTU.DK
*DTU Compute, Technical University of Denmark, Denmark*

## Abstract

We propose a semantic similarity metric for image registration. Existing metrics like Euclidean Distance or Normalized Cross-Correlation focus on aligning intensity values, giving difficulties with low intensity contrast or noise. Our approach learns dataset-specific features that drive the optimization of a learning-based registration model. We train both an unsupervised approach using an auto-encoder, and a semi-supervised approach using supplemental segmentation data to extract semantic features for image registration. Comparing to existing methods across multiple image modalities and applications, we achieve consistently high registration accuracy. A learned invariance to noise gives smoother transformations on low-quality images. Code and experiments are available at `github.com/SteffenCzolbe/DeepSimRegistration`.

**Keywords:** Image Registration, Deep Learning, Representation Learning

## 1. Introduction

Deformable registration, or nonlinear image alignment, is a fundamental tool in medical imaging. Registration models find correspondences between a set of images and derive a geometric transformation to align them. Most algorithmic and deep learning-based methods solve the registration problem by the minimization of a loss function, consisting of a similarity metric and a regularization term ensuring smoothness of the transformation. The similarity metric is essential to the optimization; it judges the quality of the match between registered images and has a strong influence on the result.

Pixel-based similarity metrics like euclidean distance and patch-wise cross-correlation are well explored within algorithmic (Avants et al., 2011, 2008; Faisal Beg et al., 2005; Rueckert et al., 1999; Thirion, 1998; Vercauteren et al., 2007) and deep learning based (Alvén et al., 2019; Balakrishnan et al., 2019; Dalca et al., 2018, 2019; de Vos et al., 2019; Liu et al., 2019; Lee et al., 2019; Hu et al., 2019a,b; Yang et al., 2017; Xu and Niethammer, 2019) image registration. These metrics assume that if the image intensities are aligned, or strongly correlated, the images are well aligned. Each choice of metric adds additional assumptions on the characteristics of the specific dataset. Thus, a common methodological approach is to trial registration models with multiple different pixel-based metrics, and choose the metric performing best on the dataset (Balakrishnan et al., 2019; Hu et al., 2019a).

The shortcomings of pixel-based similarity metrics, such as blur in the generated images, have been studied substantially in the image generation community (Hou et al., 2017; Zhang et al., 2018) and have been superseded by deep similarity metrics approximating human visual perception. Here, features are extracted from neural networks pre-trained on image-classification tasks (Deng et al., 2009). Performance can further be improved by fine-tuning the features to human perception (Czolbe et al., 2020; Zhang et al., 2018), leading to generative models that produce photo-realistic images. We propose to apply deep similarity metrics within image registration to achieve a similar increase of performance for registration models.

**Contribution**    We propose a data-driven similarity metric for image registration based on the alignment of semantic features. We explore both unsupervised (using auto-encoders) and semi-supervised methods (using a segmentation model) to learn filters of semantic importance to the dataset. We use the learned features to construct a similarity metric used for training a registration model, and validate our approach on three biomedical datasets of different image modalities and applications. Across all datasets, our method achieves consistently high registration accuracy, outperforming even metrics utilizing supervised information on two out of three datasets, while yielding an inconclusive result on the third one. Our models learn to ignore noisy image patches, leading to smoother transformations on low-quality data.

## 2. Background & related work

**Image registration**    Most image registration frameworks model the problem as finding a transformation $\Phi \colon \Omega \to \Omega$ that aligns a moving image $\mathbf{I} \colon \Omega \to \mathbb{R}$ to a fixed image $\mathbf{J} \colon \Omega \to \mathbb{R}$. The morphed source image, obtained by applying the transformation, is expressed by function composition as $\mathbf{I} \circ \Phi$. The domain $\Omega$ denotes the set of all coordinates $\mathbf{x} \in \mathbb{R}^d$ within the image[1]. Images record intensity at discrete pixel-coordinates $\mathbf{p}$ but can be viewed as a continuous function by interpolation.

The transformation is found through iterative algorithms (Avants et al., 2008; Faisal Beg et al., 2005; Rueckert et al., 1999; Thirion, 1998; Vercauteren et al., 2007), or predicted with learning based techniques (Balakrishnan et al., 2019; Dalca et al., 2018; de Vos et al., 2019; Liu et al., 2019; Lee et al., 2019; Hu et al., 2019b; Yang et al., 2017; Xu and Niethammer, 2019). In both cases, the optimal transformation is found by minimization of a similarity measure $D$ and a $\lambda$-weighted regularizer $R$, expressed via the loss function

$$L(\mathbf{I}, \mathbf{J}, \Phi) = D(\mathbf{I} \circ \Phi, \mathbf{J}) + \lambda R(\Phi) \ . \tag{1}$$

As many non-linear transformation models are over-parametrized and have multiple minima, regularization is necessary. Smooth transformation fields, that avoid folds or gaps, are assumed to be physically plausible and encouraged by the regularizer. We use the *diffusion regularizer* throughout this paper, which is defined as

$$\Phi(\mathbf{p}) = \mathbf{p} + \mathbf{u}(\mathbf{p}), \ \ R(\Phi) = \sum_{\mathbf{p} \in \Omega} \|\nabla \mathbf{u}(\mathbf{p})\|^2 \ , \tag{2}$$

with the gradient of the displacement field $\nabla \mathbf{u}(\mathbf{p})$ approximated via finite differences.

**Similarity metrics for image registration**    Denote by $D$ the (dis-)similarity between the morphed moving image $\mathbf{I} \circ \Phi$ and the fixed image $\mathbf{J}$. Pixel-based metrics are well explored within algorithmic image registration, a comparative evaluation is given by Avants et al. (2011). We briefly recall the two most popular choices, *mean squared error* (MSE) and *normalized cross correlation* (NCC). The pixel-wise MSE is intuitive and computationally efficient. It is derived from maximizing the negative log-likelihood of a Gaussian normal distribution, making it an appropriate choice under the

---

1. While the domain $\Omega$ is continuous in $\mathbb{R}^d$, recorded images and computations thereon are discrete. For simplicity of notation, we denote both the continuous and discrete domain as $\Omega$. We implement $\sum_{\mathbf{p} \in \Omega}$ as a vectorized operation over the discrete pixel/voxel-coordinates and calculate $|\Omega|$ as the total count of discrete pixels/voxels of the image. The transformation $\Phi$ is implemented as a map from a discrete domain to a continuous one, and the sampling of continuous points from a discrete image is implemented via bi-/tri-linear interpolation.

assumption of Gaussian noise. On a grid of discrete points $\mathbf{p}$ from domain $\Omega$, the MSE is defined as $\mathrm{MSE}(\mathbf{I} \circ \Phi, \mathbf{J}) = \frac{1}{|\Omega|} \sum_{\mathbf{p} \in \Omega} \|\mathbf{I} \circ \Phi(\mathbf{p}) - \mathbf{J}(\mathbf{p})\|^2$.

Patch-wise normalized cross correlation is robust to variations in brightness and contrast, making it a popular choice for images recorded with different acquisition tools and protocols, or even across image modalities. For two image patches $\mathbf{A}, \mathbf{B}$, represented as column-vectors of length $N$ with patch-wise means $\bar{\mathbf{A}}, \bar{\mathbf{B}}$ and variance $\sigma_{\mathbf{A}}^2, \sigma_{\mathbf{B}}^2$, it is defined as

$$\mathrm{NCC}_{\text{patch}}(\mathbf{A}, \mathbf{B}) = \sum_{n=1}^{N} \frac{(\mathbf{A}_n - \bar{\mathbf{A}})(\mathbf{B}_n - \bar{\mathbf{B}})}{\sigma_{\mathbf{A}} \sigma_{\mathbf{B}}} \quad . \tag{3}$$

The Patch-wise similarities are then averaged over the image (Gee et al., 1993; Avants et al., 2008). Note that an alternative, computationally efficient variant of NCC is sometimes used in image registration (Avants et al., 2011). If annotations are available, these unsupervised similarity measures can be extended by a *supervised* component to measure both intensity differences and the alignment of annotated label maps (Balakrishnan et al., 2019).

**Learned similarity metrics for image registration**  While deep-learning-based image registration has received much interest recently, similarity metrics utilizing the compositional and data-driven advantages of neural networks remain under-explored. Some current works explore how to incorporate scale-space into learned registration models, but similarity metrics remain pixel-based (Hu et al., 2019a; Li and Fan, 2018). Learned similarity metrics are proposed by Haskins et al. (2019) and Krebs et al. (2017), but both approaches require ground truth registration maps, which are either synthetically generated or manually created by a medical expert. Lee et al. (2019) propose to learn annotated structures of interest as part of the registration model to aid alignment, but the method discards sub-regional and non-annotated structures.

Learned common data representations and similarity metrics are frequently used in multi-modal image registration (Heinrich et al., 2012; Chen et al., 2016; Simonovsky et al., 2016; Pielawski et al., 2020). While these approaches learn common representations from well-aligned images of multiple modalities, we aim to find a semantically augmented representation of images of a single modality.

## 3. Method

**A discussion of NCC**  Our design of a semantic similarity metric starts by examining the popular NCC metric. We see that NCC between image patches $\mathbf{A}$ and $\mathbf{B}$ is equivalent to the cosine-similarity between the corresponding mean-centered vectors $f(\mathbf{A}) = \mathbf{A} - \bar{\mathbf{A}}$ and $f(\mathbf{B}) = \mathbf{B} - \bar{\mathbf{B}}$:

$$\mathrm{NCC}_{\text{patch}}(\mathbf{A}, \mathbf{B}) = \frac{\langle f(\mathbf{A}), f(\mathbf{B}) \rangle}{\|f(\mathbf{A})\| \|f(\mathbf{B})\|} \quad , \tag{4}$$

with scalar product $\langle \cdot, \cdot \rangle$ and euclidean norm $\| \cdot \|$. Thus, an alternative interpretation of the NCC similarity measure is the cosine-similarity between two feature descriptors in a high-dimensional space. The descriptor is given by the intensity values of a centered image patch centered at a pixel $\mathbf{p}$. We will construct a similar metric, using semantic feature descriptors instead.

**A semantic similarity metric for image registration**  To align areas of similar semantic value, we propose a similarity metric based on the agreement of semantic feature representations of two images. Semantic feature maps are obtained by a *feature extractor*, which is pre-trained on a

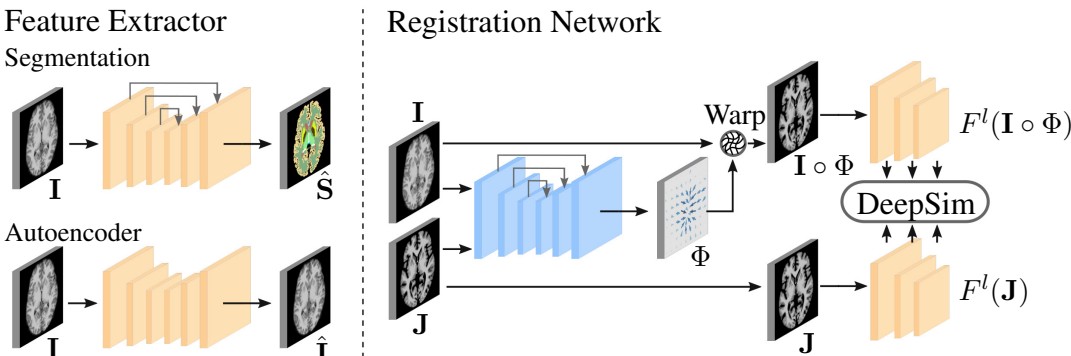

Figure 1: Two-step training: First, the feature extractor (yellow) is trained. We test both a segmentation model, and an auto-encoder. Next, the feature extractor weights are frozen and used in the loss computation of the registration network (blue).

surrogate task. To capture alignment of both localized, concrete features, and global, abstract ones, we calculate the similarity at multiple layers of abstraction. Given a set of feature-extracting functions $F^l \colon \mathbb{R}^{\Omega \times C} \to \mathbb{R}^{\Omega_l \times C_l}$ for $L$ layers, we define

$$\mathrm{DeepSim}(\mathbf{I} \circ \Phi, \mathbf{J}) = \frac{1}{L} \sum_{l=1}^{L} \frac{1}{|\Omega_l|} \sum_{\mathbf{p} \in \Omega_l} \frac{\langle F_{\mathbf{p}}^l(\mathbf{I} \circ \Phi), F_{\mathbf{p}}^l(\mathbf{J}) \rangle}{\|F_{\mathbf{p}}^l(\mathbf{I} \circ \Phi)\| \|F_{\mathbf{p}}^l(\mathbf{J})\|} \ , \tag{5}$$

where $F_{\mathbf{p}}^l(\mathbf{J})$ denotes the $l^{th}$ layer feature extractor applied to image $\mathbf{J}$, at spatial coordinate $\mathbf{p}$. It is a vector of $C_l$ output channels, and the spatial size of the $l^{th}$ feature map is denoted by $|\Omega_l|$.

Similarly to NCC, the neighborhood of the pixel is considered in the metric, as we compose $F^l$ of convolutional filters with increasing receptive area of the composition. In contrast, it is not necessary to zero-mean the feature descriptors, as the semantic feature representations are trained to be robust to variances in image brightness present in the training data.

**Feature extraction** To aid registration, the functions $F^l(\cdot)$ should extract features of semantic relevance for the registration task, while ignoring noise and artifacts. We extract features from the encoding branch of networks trained on two surrogate tasks.

First, if segmentation masks are available, we can learn features on a supplementary segmentation task. Segmentation models excel at learning relevant kernels for the data while attaining invariance towards non-predictive features like noise, but require an annotated dataset for training. We denote the proposed similarity metric with feature extractors conditioned on this task as $\mathrm{DeepSim}_{\mathrm{seg}}$.

Second, we can learn an abstract feature representation of the dataset in an unsupervised setting with auto-encoders. Auto-encoders learn efficient data encoding by training the network to ignore signal noise. A benefit of this approach is that no additional annotations are required. While variational methods for encoding tasks have several advantages, we choose a deterministic auto-encoder for its simplicity and lack of hyperparameters. We denote the similarity metric with feature extractors conditioned on this task as $\mathrm{DeepSim}_{\mathrm{ae}}$.

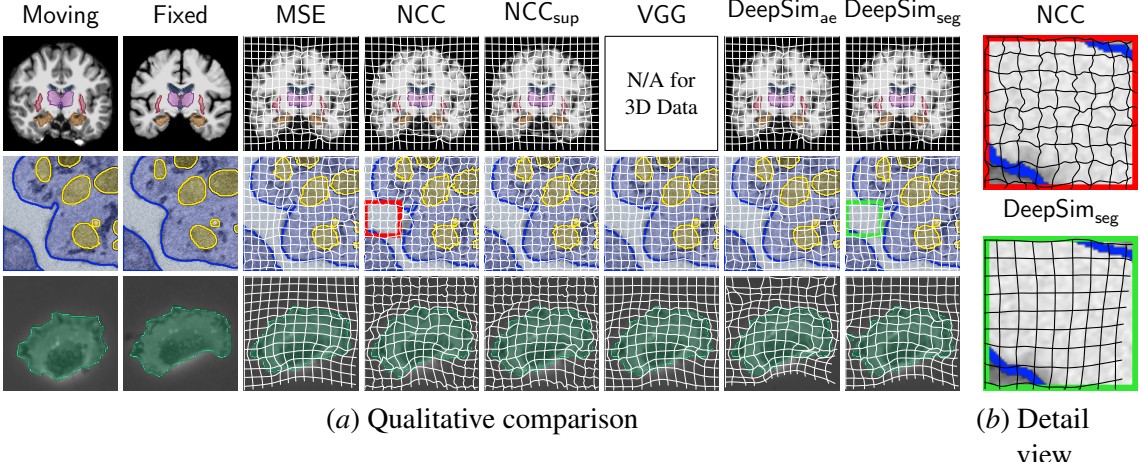

(a) Qualitative comparison  (b) Detail view

Figure 2: Left: Qualitative comparison of registration models. Rows: Datasets Brain-MRI, Platelet-EM, PhC-U373. Columns 3-8: Registration models trained with various similarity metrics. Right: Detail view of highlighted noisy background areas on the Platelet-EM dataset. Select segmentation classes annotated in color. The transformation is visualized by morphed grid-lines.

## 4. Experiments

We empirically compare registration models trained with the unsupervised $\mathrm{DeepSim_{ae}}$ and semi-supervised $\mathrm{DeepSim_{seg}}$ to the baselines MSE, NCC, $\mathrm{NCC_{sup}}$ (NCC with supervised information), and VGG (a VGG-net based deep similarity metric from image generation). Our implementation of the baseline methods follows Avants et al. (2011), Balakrishnan et al. (2019), and Hou et al. (2017).

As our goal is not to advance the state of the art for any particular registration task, but instead to explore the value of our loss-function in a generic setting, we use well-established 2D

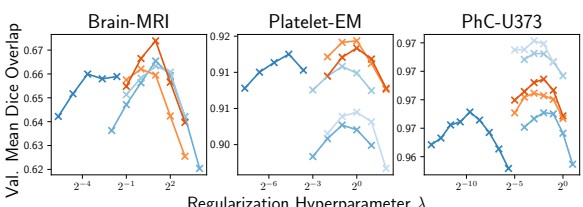

Figure 3: Hyperparameter tuning. For each model, we select the regularizer weighting factor $\lambda$ with the highest validation mean dice overlap for further evaluation. Color scheme follows Figure 4.

and 3D U-Net (Ronneberger et al., 2015) architectures for both registration and segmentation models, sketched in Figure 1. For the auto-encoder task, we use the same architecture, but without the shortcut connections. Each network consists of three encoder and decoder stages. Each stage consists of one batch normalization (Ioffe and Szegedy, 2015), two convolutional, and one dropout layer (Gal, Yarin and Ghahramani, 2016). After the final decoder step, we smooth the model output with three more convolutional layers. We experimented with deeper architectures but found they do not increase performance. The activation function is LeakyReLu throughout the network, Softmax for the final layer of the segmentation network, Sigmoid for the final layer of the auto-encoder, and linear for the final layer of the registration network. The stages have $64, 128, 256$ channels for 2d datasets, and $32, 64, 128$ channels for 3d.

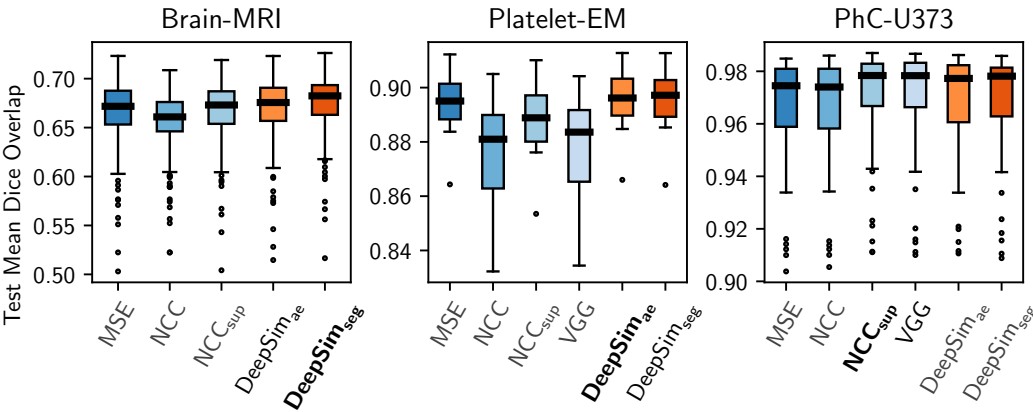

Figure 4: Mean dice overlap of registration models on the test set. Baselines in shades of blue, ours in red. Label of the best metric in bold, 2nd best black, others in grey. Boxplot with median, quartiles, deciles and outliers.

The segmentation model is trained with a cross-entropy loss function, the auto-encoder with the mean squared error, and the registration network with the loss given by Eq. 1. The optimization algorithm for all models is ADAM (Kingma and Ba, 2015), the initial learning rate is $10^{-4}$, decreasing by a factor of 10 each time the validation loss plateaus. All models are trained until convergence. Due to the large 3D volumes involved, the choice of batch-size is often limited by available memory. We sum gradients over multiple passes to arrive at effective batch-sizes of 3-5 samples. The hyperparameter $\lambda$ is tuned for each model independently on the validation set, we plot the validation score of tested values in Figure 3.

To show that our approach applies to a variety of registration tasks, we validate it on three 2D and 3D datasets of different modalities: 4000 T1-weighted *Brain-MRI* scans (Di Martino et al., 2014; LaMontagne et al., 2019), 74 slices of human blood cells of the *Platelet-EM* dataset (Quay et al., 2018), and 230 time-steps of the cell-tracking video *PhC-U373* (Maška et al., 2014; Ulman et al., 2017). All datasets are pre-aligned through affine transformations and split into train, validation, and test sections. We augment each pair of images with random affine transformations during training.

## 5. Results

**Registration accuracy**   We measure the mean Sørensen Dice coefficient on the unseen test-set in Figure 4. Statistical significance testing of the results is performed with the Wilcoxon signed rank test for paired samples. A significance level of 5% gives Bonferroni-adjusted significance threshold $p = 0.002$. We further measure the effect size with Cohen's d and show the results in Table 1. Models trained with our proposed $\mathrm{DeepSim_{ae}}$ and $\mathrm{DeepSim_{seg}}$ outperform all baselines on the Brain-MRI and Platelet-EM datasets, with strong statistical significance and effect size. On the PhC-U373 dataset, all models achieve high dice-overlaps of $> 0.97$.

We monitor the mean dice overlap during training. The training accuracy is, with few exceptions, similar to the test accuracy, indicating that results generalize well. The empirical convergence speeds of the tested metrics differ. We observe that the $\mathrm{DeepSim_{seg}}$ converges faster than the baselines, especially in the first few epochs of training. See appendix A for a convergence plot.

| Dataset | Method | Baseline | | | | | | | |
|---------|--------|----------|---|----------|---|----------|---|----------|---|
| | | MSE | | NCC | | NCC$_{\text{sup}}$ | | VGG | |
| | | $p$ | $d$ | $p$ | $d$ | $p$ | $d$ | $p$ | $d$ |
| Brain-MRI | DeepSim$_{\text{ae}}$ | $<0.001$ | 0.14 | $<0.001$ | 0.43 | $<0.001$ | 0.10 | – | – |
| | DeepSim$_{\text{seg}}$ | $<0.001$ | 0.30 | $<0.001$ | 0.60 | $<0.001$ | 0.25 | – | – |
| Platelet-EM | DeepSim$_{\text{ae}}$ | 0.016 | 0.12 | $<0.001$ | 1.14 | $<0.001$ | 0.51 | $<0.001$ | 1.06 |
| | DeepSim$_{\text{seg}}$ | 0.034 | 0.10 | $<0.001$ | 1.12 | $<0.001$ | 0.49 | $<0.001$ | 1.04 |
| PhC-U373 | DeepSim$_{\text{ae}}$ | $<0.001$ | 0.10 | $<0.001$ | 0.11 | $<0.001$ | $-0.06$ | $<0.001$ | $-0.05$ |
| | DeepSim$_{\text{seg}}$ | $<0.001$ | 0.12 | $<0.001$ | 0.13 | 0.002 | $-0.03$ | 0.003 | $-0.02$ |

Table 1: Results of the statistical significance test, performed with the Wilcoxon signed rank test for paired samples. Effect size measured with Cohen's d. Statistically insignificant results (significance level 0.05, Bonferroni-corrected to $p > 0.002$) and very small effect sizes ($|d| < 0.1$) in grey.

**Qualitative examples & transformation grids**   We plot the fixed and moving images $\mathbf{I}$, $\mathbf{J}$ and the morphed image $\mathbf{I} \circ \Phi$ for each similarity metric model along with a more detailed view of a noisy patch of the Platelet-EM dataset in Figure 2, and perform a quantitative analysis of the transformation in Appendix B. On models trained with the baselines, we find strongly distorted transformation fields in noisy areas of the images. In particular, models trained with NCC and NCC$_{\text{sup}}$ produce very irregular transformations, despite careful tuning of the regularization-hyper-parameter. The model trained by DeepSim$_{\text{seg}}$ is more invariant towards noise.

**Anatomical regions**   The Brain-MRI dataset is annotated with the anatomical regions of the brain. We plot the dice overlap per region in a boxplot in Figure 5, and highlight regions where both of our metrics perform better than all baselines bold. Baseline methods (blue) perform very similar, despite NCC$_{\text{sup}}$ as a supervised metric requiring more information over the unsupervised MSE and NCC.

## 6. Discussion & Conclusion

Registration models trained with DeepSim achieve high registration accuracy across datasets, allowing improved downstream analysis and diagnosis. Its consistency makes testing multiple traditional metrics unnecessary; instead of empirically determining whether MSE or NCC captures the characteristics of a data-set best, we can use DeepSim to learn the relevant features from the data.

Our experiments show that the availability of annotated anatomical regions can help in learning semantic features, but is not a necessity. Gains in performance in deep learning often require large, annotated datasets, which are expensive and time-consuming to obtain in biomedical settings. The feature extractor of DeepSim$_{\text{ae}}$ was trained on the unsupervised autoencoding task, requiring no additional data aside from the intensity images to be registered. It performed similarly to the semi-supervised DeepSim$_{\text{seg}}$ on two of the three datasets and outperformed the baselines.

The analysis of noisy patches in Figure 2 and Appendix B highlights a learned invariance to noise. The pixel-based similarity metrics are distracted by artifacts, leading to overly-detailed transformation fields. Models trained with DeepSim do not show this problem. While smoother transformation fields can be obtained for all metrics by strengthening the regularizer, this would

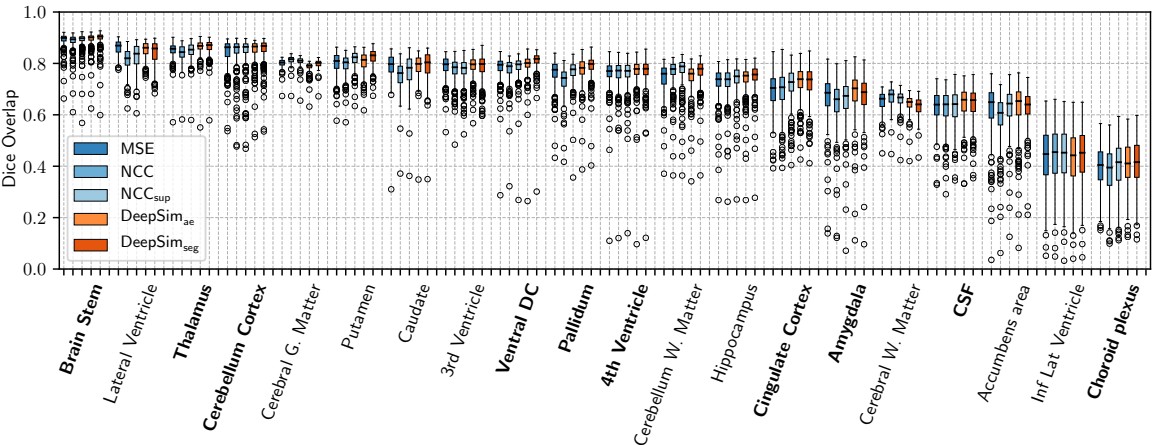

Figure 5: Dice overlaps of the anatomical regions of the Brain-MRI dataset. Baselines in shades of blue, our methods in red. Bold labels for regions where *both* of our methods score higher than all baselines. We combined labels of the left and right brain hemispheres into a single class. The boxplot shows median, quartiles, deciles and outliers.

negatively impact the registration accuracy of anatomically significant regions. Accurate registration of noisy, low-quality images allows for shorter acquisition time and reduced radiation dose in medical applications.

A weakness of DeepSim is the need to train a separate model for feature extraction. The design, training, and testing of a second model takes additional resources, and the presented approach necessitates a dataset to train the feature extractor with. In the context of deep learning-based registration, DeepSim$_{ae}$ requires no additional data to what is required to train the registration model, while DeepSim$_{seg}$ requires additional label maps. Compared to algorithmic registration methods, which optimize the registration map for each pair of input images separately, any deep-learning approach requires additional data.

DeepSim is a general metric, applicable to image registration tasks of all modalities and anatomies. Beyond the presented datasets, our good results in the presence of noise let us hope that DeepSim will improve registration accuracy in domains such as low dose CT, ultrasound, or microscopy, where details are often hard to identify, and image quality is poor. We further emphasize that the application of DeepSim is limited neither to deep learning nor to image registration. In Algorithmic image registration, a similarity-based loss is minimized via gradient descent-based methods. DeepSim can be applied to drive algorithmic methods, improving their performance by aligning deep, semantic feature embeddings. Similarly, DeepSim could be used for other image regression tasks, such as image synthesis, -translation, or -reconstruction.

## Acknowledgments

This work was funded in part by the Novo Nordisk Foundation through the Center for Basic Machine Learning Research in Life Science (grant no. 0062606), and in part through the Lundbeck Foundation (grant no. R218-2016-883). We further thank Matthew Quay, the Cell Tracking Challenge, and the Cancer Imaging Archive for the provision of the datasets.

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

## Appendix A. Optimization convergence

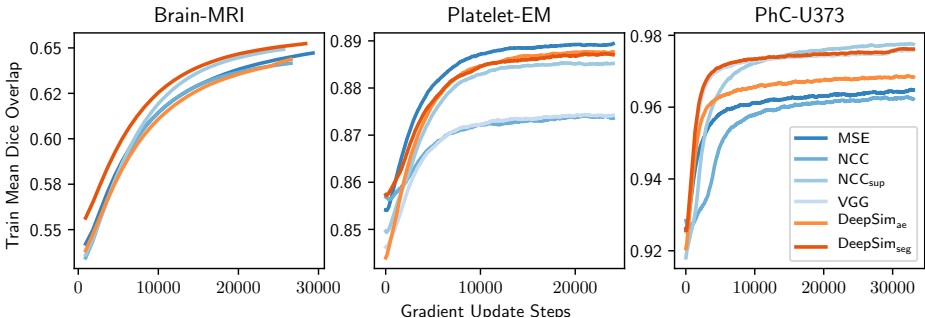

Figure 6: Mean dice overlap of the registration models during training. The training duration per model on a single RTX 2080 GPU is approximately seven days for the Brain-MRI dataset and one day for the Platelet, PhC-U373 datasets.

## Appendix B. Regularity of the transformation

| Method | Dataset | | | | | |
|---|---|---|---|---|---|---|
| | Brain-MRI | | Platelet-EM | | PhC-U373 | |
| | $\sigma^2(J_\Phi)$ | $|J_\Phi| < 0\,(\%)$ | $\sigma^2(J_\Phi)$ | $|J_\Phi| < 0\,(\%)$ | $\sigma^2(J_\Phi)$ | $|J_\Phi| < 0\,(\%)$ |
| MSE | 0.19 | 0.42 | 0.23 | 0.40 | 0.03 | 0.02 |
| NCC | 0.23 | 0.93 | 0.54 | 4.15 | 0.29 | 0.71 |
| NCC$_{sup}$ | 0.11 | 0.28 | 0.54 | 4.03 | 0.28 | 0.57 |
| VGG | – | – | 0.09 | 0.00 | 0.16 | 0.13 |
| DeepSim$_{ae}$ | 0.13 | 0.20 | 0.10 | 0.04 | 0.20 | 0.35 |
| DeepSim$_{seg}$ | 0.09 | 0.12 | 0.13 | 0.14 | 0.18 | 0.32 |

Table 2: Regularity of the transformation. The determinant of the Jacobian of the transformation $|J_\Phi|$ is a measure of how the image volume is compressed or stretched by the transformation. We assess transformation smoothness by the variance of the voxel-wise determinant $\sigma^2(J_\Phi)$, a lower variance indicates a more volume-preserving transformation. We assess the regularity of the transformation by measuring the percentage of voxels for which the determinant is $< 0$, which indicates domain folding.

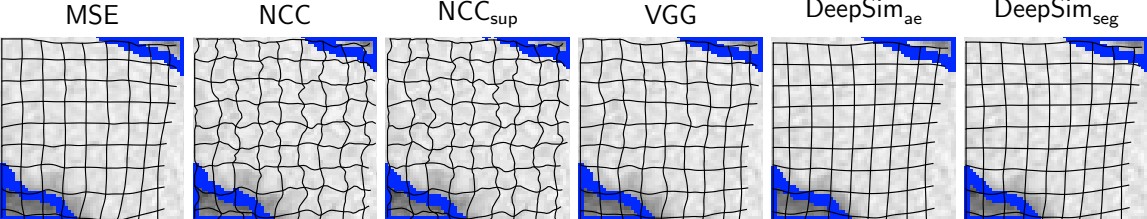

Figure 7: Addendum to Figure 2(*b*). Detail view of noisy background areas on the Platelet-EM dataset. The cell-boundary is annotated in blue. The transformation is visualized by morphed grid-lines.

