# OpenReview forum: "Semantic similarity metrics for learned image registration"
_MIDL.io/2021/Conference — MIDL 2021_

### Official Review · AnonReviewer3 · 2021-02-28

**Confidence:** 3
**Preliminary Rating:** 4
**Recommendation:** Oral

**Summary:**

The authors introduce a novel data-driven similarity metric for image registration by leveraging the alignment of semantic features.  They used both unsupervised and semi-supervised methods to learn filters of semantic features then they leveraged the resulting features to construct a similarity metric used for training a registration model. Extensive results show the importance of their proposed solution compared with the benchmark methods.

**Strengths:**

Results are clear and do a good job of emphasizing better results with proposed idea. Statistical tests in the paper are good support material for significance of results. Mathematical definitions are well articulated and well linked int he paper.

**Weaknesses:**

Different from the experimental results that were well displayed and expressed, still the discussion section poorly developped in terms of the limitations of the work in detail from a deep learning perspective not a clinical level. Figure 5 is very small with crowded plots. The section 2 is very detailed while the section 3 should more detailed than 2.

**Deanonymize Review:**

no

**Justification Of The Preliminary Rating:**

I see a professional writing style and a professional deep learning framework design. The authors explained well the method section and reported significant results to demonstrated the potential of their work. The English writing is good as well as the figure design of the main architecture.

**Paper Type:**

methodological development

**Special Issue:**

yes

---

> ### Author Response · Authors · 2021-03-17
> **Response to Reviewer 3**
>
> Thank you for the review.
>
> **On weaknesses / critical discussion:** We included the following paragraph in the discussion (see updated submission):
>
> > A weakness of DeepSim is the need to train a separate model for feature extraction. The design, training, and testing of a second model takes additional resources, and the presented approach necessitates a dataset to train the feature extractor with. In the context of deep learning-based registration, DeepSim_ae requires no additional data to what is required to train the registration model, while DeepSim_seg requires additional label maps. Compared to algorithmic registration methods, which optimize the registration map for each pair of input images separately, any deep-learning approach requires additional data.
>
> **On Figure legibility:** The presented plots are best viewed on a digital display. Note that all figures are embedded as vector graphics, allowing infinite zoom. Based on your comment, we did adjust the line width and borders to be less intrusive in Fig. 5 in updated submission.

---

### Official Review · ~Alessa_Hering1 · 2021-03-06

**Confidence:** 4
**Preliminary Rating:** 3
**Recommendation:** Oral, Poster
**Final Rating:** 4

**Summary:**

The manuscript presents a method to learn a similarity metric for image registration from a dataset. A network for a surrogate task (two different ones are tested in this paper: a segmentation network and an auto-encoder) are trained and the features from the encoding branch are used as features for the registration. The method is evaluated on three different tasks (brain MRI, slices of human blood cells, cell tracking video)

**Strengths:**

In general, I enjoyed reading the manuscript. It is well structured, has a clear motivation, and is well written.
- The authors clearly point out that their goal is not to achieve state of the art registration results but explore the value of different loss functions
- mathematical formulation of the problem
- comparison to several other similarity metrics (MSE, NCC; NCC + supervision, VGG)
- the experiments are performed on three tasks (brain MRI, slices of human blood cells, cell tracking video) which are very diverse
- statistical tests were performed
- clear and informative visualization of the results
- code is available


**Weaknesses:**

- missing reference regarding image descriptors for image registration, e.g.
      - Heinrich, M. P., Jenkinson, M., Bhushan, M., Matin, T., Gleeson, F. V., Brady, M., & Schnabel, J. A. (2012). MIND: Modality independent neighborhood descriptor for multi-modal deformable registration. Medical image analysis, 16(7), 1423-1435.


- Missing references regarding other learned similarity metrics, e.g.
    -  Blendowski, M., Heinrich, M.P.: Combining MRF-based deformable registration and deep binary 3D-CNN descriptors for large lung motion estimation in COPD patients. Int. J. Comput. Assisted Radiol. Surg. 14, 1–10 (2018)
  - Niethammer, M., Kwitt, R., & Vialard, F. X. (2019). Metric learning for image registration. In Proceedings of the IEEE/CVF Conference on Computer Vision and Pattern Recognition (pp. 8463-8472).
  - Cheng, X., Zhang, L., Zheng,Y.: Deep similarity learning for multimodal medical images. Comput. Methods Biomech. Biomed. Eng.
  - Krebs, J., Mansi, T., Delingette, H., Zhang, L., Ghesu, F.C., Miao, S., Maier, A.K., Ayache, N., Liao, R., Kamen, A.: Robust non-rigid registration through agent-based action learning. In: International Conference on Medical Image Computing and Computer-Assisted Intervention, pp. 344–352. Springer, Berlin (2017)
  - Simonovsky, M., Gutiérrez-Becker, B., Mateus, D., Navab, N., Komodakis, N.: A deep metric for multimodal registration. In: International Conference on Medical Image Computing and Computer-Assisted Intervention, pp. 10–18. Springer, Berlin (2016)

-  While the discussion goes into detail about advantages and possible areas of application, a critical examination of the results is missing. Are there any limitations of this approach (e.g. a certain number of available images)?

Open Questions:
- Does the choice of the similarity measure have an impact on the regularity of the deformation field?


**Deanonymize Review:**

yes

**Final Rating Justification:**

I appreciate the updates the author made and with them, the presented paper is even stronger.

**Justification Of The Preliminary Rating:**

The authors present an interesting new method for learning features for image registration. The paper is well structured and written. However, a few references are missing and the discussion could be more critical.

**Paper Type:**

methodological development

**Questions To Address In The Rebuttal:**

I already like your work like it is and it could be accepted. Nevertheless, there are some points that would improve your work and should be addressed in the rebuttal.

-	Please add and discuss some of the mentioned references. What is the advantage/drawback of the presented approach?
-	What are the weaknesses of your approach? Are there any cases in which your method failed? (You could add a few more example images in the appendix)
-	Please add an analysis regarding the regularity of the deformation field for the different similarity measures.


**Special Issue:**

no

---

> ### Author Response · Authors · 2021-03-17
> **Response to Reviewer 2**
>
> Thank you for the review and the detailed list of relevant related works.
>
> **On the named papers:** Thanks for the literature suggestions. Some of them are indeed relevant to our work. We include a reference to similarity metrics for multimodal image registration in the updated submission. We also summarize their relation to our work below, citing them as [1] ... [6] in the order of your comment:
>
> * [1,4,6] are learning features based on obtaining feature representations using different image modalities, but in our case, we are only looking at a single modality.
> * [2, 5] use additional spatial information like key points or synthetic registration maps, which are not available in our application.
> * [3] adapt the regularizer based on the input images without changing the similarity metric.
>
> We see our work as adding on top of this prior work in extracting more information from a single modality, and thus we think that it can be worthwhile to combine it with the mentioned approaches.
>
>
> **On weaknesses:** We included the following paragraph in the discussion (see updated submission):
>
> > A weakness of DeepSim is the need to train a separate model for feature extraction. The design, training, and testing of a second model take additional resources, and the presented approach necessitates a dataset to train the feature extractor with. In the context of deep learning-based registration, DeepSim_ae requires no additional data to what is required to train the registration model, while DeepSim_seg requires additional label maps. Compared to algorithmic registration methods, which optimize the registration map for each pair of input images separately, any deep-learning approach requires additional data.
>
> **On regularity of the deformation field:** We added a detailed analysis of the transformation in Appendix B (see updated submission). Citing from the updated paper:
>
> > The Jacobian of the transformation $|J_{\Phi}|$ is a measure of how the image volume is compressed or stretched by the transformation. We assess transformation smoothness by the variance of the voxel-wise log absolute determinant $\sigma^2(\log |J_{\Phi}|)$, a lower variance indicates a more volume-preserving transformation. We assess the regularity of the transformation by measuring the percentage of voxels for which the determinant is < 0, which indicates domain folding.
>
> In Fig. 3(b), we compared the transformation grids of our method DeepSim_seg to the baseline NCC on a noisy image patch. Due to space constraints, it was not possible to include the same image for all metrics. We now added these missing images to Appendix B.
>
> The additional qualitative and quantitative results support our findings.

---

> > ### Comment · AnonReviewer2 · 2021-03-21
> >
> > Thank you for the detailed answer.
> >
> > I appreciate the updates the author made and with them, the presented paper is even stronger. I am looking forward to the presentation at MIDL.

---

### Official Review · AnonReviewer1 · 2021-03-09

**Confidence:** 5
**Preliminary Rating:** 4
**Recommendation:** Oral

**Summary:**

This work proposed a data-driven semantic similarity metric for image registration. Instead of Euclidean distance or normalised cross-correlation, which focus on aligning intensity values, the proposed metric focuses on dataset-specific semantic feature extracted though deep learning. From the interpretation of well-known normalised cross-correlation, the authors introduced cosine similarity, and extend it as deep similarity metric. The experimental evaluation including the comparison with existing metrics for three different dataset showed the validity of the proposed metric. The registration by the trained model with their proposed metric achieved consistently high registration accuracy for the three datasets.

**Strengths:**

As follows:
1)	Paper is well structure and solid work.
2)	Surveyed is satisfactory, and backgrounds are clear.
3)	Description about the proposed method is concise and easy to grasp.
4)	Experiments with three different kind dataset showed the validity of the proposed method via qualitative comparison and quantitative evaluation including statistical evaluation metric.
5)	Experimental results clarified that the availability of annotated anatomical regions can help in learning semantic features, but is not a necessary in their framework. This is important remark of this work.

**Weaknesses:**

Even though authors present experimental evaluations to show the validity of the proposed similarity metric, why the proposed method is theoretically good is still ambiguous for me. Semantic feature extraction mitigates the noise effects; yes, it contributes to accurate registration. However, why cosine similarity between extracted semantic features is good for registration is still unclear. For classical feature that directory expresses an intensity distribution of local patch, cosine similarity is robust against illumination changes and small pattern perturbations. For semantic features, what is the advantage of cosine similarity?

**Deanonymize Review:**

no

**Detailed Comments:**

Minors: Abstract, euclidean distance=> Euclidean distance

**Justification Of The Preliminary Rating:**

For image registration, this work proposed a new data-driven semantic similarity metric as a solid work. How solid is summarised in the list at Strengths. Experimental evaluation and discussion are convincing. Furthermore, this method has potential demands for many applications as preprocessing, since it does not require annotation labels of organ regions.

**Paper Type:**

methodological development

**Questions To Address In The Rebuttal:**

I still have a question about why average of cosine similarity between semantic features over several layers is powerful. I think that the similarities over several layers measure the similarity in many scales, therefore, it makes sense. However, the contribution of cosine similarity itself is still unclear. If we assume that the pattern distribution on the unit sphere in high-dimensional feature space, where patterns directly introduced from intensity distributions on patches distributed, cosine similarity have similar characteristic to a geodesic distance on the unit sphere(if the distance between two given feature vectors on a unit sphere  is small,  an angle between the two feature vectors is a kind of geodesic distance and angle is defined as the arccosine of cosine similarity). Therefore, a geodesic distance is robust against illumination changes and small pattern perturbations. However, we have semantic features. In the case of the semantic feature, why a cosine similarity is good? or how a cosine similarity works?

**Special Issue:**

yes

---

> ### Author Response · Authors · 2021-03-17
> **Response to Reviewer 1**
>
> Thank you for the thorough feedback.
>
> **About averaging the metric across multiple layers:** As you point out, this is effectively building a feature pyramid, measuring the similarity at multiple scales. In addition, network activations of different layers of a NN represent the data at different levels of abstraction.
>
> **Concerning the choice and suitability of cosine similarity:** We chose cosine similarity because it follows from NCC, which inspired the approach presented here. The normalization property of cosine similarity helps to both normalize input-vectors and adds robustness towards outlier features. The first property makes it easier to combine features from different layers, which might have very different scales, while the latter is a desired feature since the underlying neural network was not trained to produce feature values in a given interval. Other metrics for comparing the semantic representations might work as well, we have not investigated them further and leave it to future work.

---

### Meta-Review · Area_Chair1 · 2021-03-29

**Recommendation:** Accept (Oral & Special Issue Candidate)

**Metareview:**

The paper presents a similarity metric for image registration based on a cosine distance between semantic features at different scales. Two options: unsupervised and supervised are considered. These propositions bring novel considerations to the problem of deep registration.  The three reviewers agree the approach is clearly exposed, and on the solidity of the experimental validation on three datasets with numerous comparisons. R1 raised a question on the theory justifying the choices and the results, the rebuttal provided some initial answers that could be added to the paper. R2 suggested adding missing relevant work, mentioning the work limitations, and analyzing the effect of the similarity on the regularity of the deformation field, all of which were addressed in the rebuttal and revised paper.

**Paper Type:**

methodological development

---

### Decision · Program_Chairs · 2021-03-31

**Decision:**

Accept

**Comment:**

Congratulations your paper has been selected as a long oral.